# TUSK: Task-Agnostic Unsupervised Keypoints

**Yuhe Jin**[1], **Weiwei Sun**[1], **Jan Hosang**[2], **Eduard Trulls**[2], **Kwang Moo Yi**[1]

[1]The University of British Columbia,   [2]Google Research

## Abstract

Existing unsupervised methods for keypoint learning rely heavily on the assumption that a specific keypoint type (*e.g.* elbow, digit, abstract geometric shape) appears only once in an image. This greatly limits their applicability, as each instance must be isolated before applying the method—an issue that is never discussed or evaluated. We thus propose a novel method to learn **T**ask-agnostic, **U**n**S**upervised **K**eypoints (TUSK) which can deal with multiple instances. To achieve this, instead of the commonly-used strategy of detecting multiple heatmaps, each dedicated to a specific keypoint type, we use a *single* heatmap for detection, and enable unsupervised learning of keypoint types through clustering. Specifically, we encode semantics into the keypoints by teaching them to reconstruct images from a sparse set of keypoints and their descriptors, where the descriptors are forced to form distinct clusters in feature space around learned *prototypes*. This makes our approach amenable to a wider range of tasks than any previous unsupervised keypoint method: we show experiments on multiple-instance detection and classification, object discovery, and landmark detection—all unsupervised—with performance on par with the state of the art, while also being able to deal with multiple instances.

## 1 Introduction

Keypoint-based methods are a popular off-the-shelf building block for a wide array of computer vision tasks, including city-scale 3D mapping and re-localization [39, 52], landmark detection on human faces [53, 64, 23, 37], human pose detection [53, 64, 37], object detection [38, 32, 13, 46], scene categorization [31, 15], image retrieval [26, 44], robot navigation [42, 12], and many others. As elsewhere in computer vision, learning-based strategies have become the de-facto standard in keypoint detection and description [53, 7, 44, 64, 23, 9, 37, 55]. This is accomplished by tailoring a method to a specific task by means of domain-specific supervision signals—methods are often simply not applicable outside the use case they were trained for. This is problematic, as many applications require extensive amounts of labeled training data, which is often difficult or infeasible to collect.

Researchers have attempted to overcome the requirement for labelled data through unsupervised or self-supervised learning [53, 64, 23, 37, 7, 48, 62], primarily using two families of techniques, often in combination: (1) enforcing equivariant representations through synthetic image transformations, such as homographies [7, 48] or non-rigid deformations [28, 53]; and (2) clustering features in a latent space over a predetermined number of channels, where every channel is meant to represent *a single object* or *a single part* of an object. The first objective ensures that keypoints can deal with changes due to camera pose or body articulation, but synthetic transformations do not capture any high-level semantics. The second objective can encode semantic information into the keypoints, but methods assume that every keypoint (a specific *object* or *part*, such as the corner of the mouth) appears only once in a given image, such that a 'specialized' heatmap can be learned for each semantic keypoint. This is an effective way to constrain optimization, but is unrealistic in many scenarios, and restricts the applicability of these methods to isolated instances—for example, a learned landmark detector for

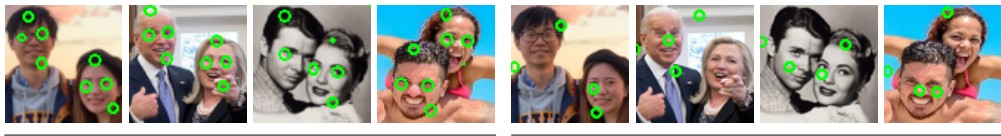

| Ours | LatentKeypointGAN [19] |

Figure 1: **Landmark detection on images with multiple faces. (Left)** Our model is trained on CelebA [35], which contains single faces, but can be trivially applied here by simply increasing the number of keypoints. **(Right)** Methods based on the multi-channel-detection paradigm fail to generalize, and would require processing each instance separately . Images are in the public domain.

human faces will break down if we simply introduce a second face to the image: see Fig. 1. In other words, these methods require each object instance to be isolated before keypoint extraction—which is itself a non-trivial problem, and makes them unsuitable as a generic keypoint detector.

We propose a method to solve this. Similarly to previous works, we estimate a **heatmap** representing the likelihood of each pixel being a keypoint, and a **feature map** encoding a descriptor for each pixel, should that pixel be a keypoint. To avoid being limited to single instances, unlike most existing works [53, 64, 23, 37, 19], we predict a *single heatmap*, from which we extract multiple keypoints. We then focus on the properties that we *want* keypoints to have, and show how to accomplish them without supervision or task-specific assumptions. Specifically, we focus on the two key aspects outlined above: (a) keypoints should be equivariant to **geometric** transformations, *i.e.*, they should follow the same transformation that the image does; and (b) group semantically similar things together. The former can be easily achieved by enforcing equivariance of both keypoint locations and their descriptors under random perturbations, following previous efforts like [53]—with additional considerations, because, unlike existing methods, we do not assume that the images contain a single instance.

The latter is more difficult to solve under the multiple-instance scenario. We address it with a novel solution based on unsupervised clustering of the feature space. We learn a latent representation for each keypoint that we use to reconstruct the image, similarly to [64, 23, 37, 16, 3, 36], and show how to discover the re-occurring structure or semantic information by clustering these features in the latent space. Specifically, we learn a *codebook* of cluster centers—which we call **prototypes**—and enforce that the distribution of descriptors at keypoint locations corresponds to the distribution of these prototypes through a novel sliced Wasserstein objective [6] that utilizes Gaussian Mixture Models (GMM). This naturally leads to descriptors clustering around the prototypes.

Our approach allows us to achieve both objectives without task-specific labels, including implicit assumptions such as that the images contain the same *objects*, or that each *object* appears only once [53, 64, 23, 37, 19]: see Fig. 1 for examples. Our main contributions are as follows:

- We show how to encode both *geometry* and *semantics* into keypoints and descriptors, without supervision. For the latter, we propose a novel technique based on prototype clustering.
- Unlike existing methods, our method does not require knowing the exact number of keypoints a prior, and can naturally deal with multiple instances and occlusions.
- Our approach does not require domain-specific knowledge and can be applied to multiple tasks— we showcase landmark detection, multiple-instance classification, and object discovery, with performance on part with the state of the art—while being able to deal with multiple instances. Other than hyperparameters, the only component we tailor to the task is the loss used to reconstruct the image from the keypoints ($\mathcal{L}_{\text{recon}}$, Sec. 3.3). Our method is the only one applicable to all tasks.

## 2 Related work

**Landmark learning**. The term *landmark* is used in the literature to refer to localized, compact representations of object structures such as human body parts. We favor the term *keypoint* instead, as our method can be applied to a variety of tasks, and our keypoints may represent both whole objects or their parts. There is a vast amount of literature on this topic, especially for human bodies [50, 49, 43] and faces [59, 65], the majority of which use manual annotations for supervision. Among unsupervised methods, Thewlis *et al*. [53] enforce landmarks to be equivariant by training a siamese network with synthetic transformations of the input image. They introduce a diversity loss that penalizes overlapping landmark distributions to prevent different channels from collapsing into the same landmark. In a follow up work, Thewlis *et al*. [54] further improve this idea by

enforcing equivariance on a dense model, rather than sparse landmarks. Zhang *et al.* [64] add a reconstruction task solved with an autoencoder, in addition to the equivariance constraint, which imbues the landmarks with semantics but still relies on separation constraints to prevent landmark collapse. Our approach does not require such an additional regularization, as our formulation prevents collapse by design. Given a pair of source and target images, Jakab *et al.* [23] detect landmarks in the target image and combine them with a global representation of the source image to reconstruct the target, and learn without supervision through either synthetic deformations or neighboring frames from video sequences. Lorenz *et al.* [37] use a two-stream autoencoder to disentangle shape and appearance and learn to reconstruct an object from its parts. We use similar strategies to enforce equivariance in the location of the landmarks (keypoints), but additionally learn distinctive features with semantic meaning. In concurrent work, He *et al.* [19] pushed performance in unsupervised keypoint detection further with a GAN-based framework. All of these methods, however, use dedicated heatmap channels for each landmark to be detected and thus assume that the image contains *one and only one object instance*—by contrast, our approach learns a single-channel heatmap and can handle a variable number of objects (see Fig. 1 for a comparison with [19]).

**Unsupervised object discovery**. Our method can be applied to object discovery, by representing *objects* as *keypoints*. Decomposing complex visual scenes with a variable number of potentially repeating and partially occluded objects into semantically meaningful components is a fundamental cognitive ability, and remains an open challenge. Finding objects in images has been historically addressed through bounding-box-based detection or segmentation methods trained on manual annotations [18, 51, 14]—learning object-centric representations of scenes without supervision has only recently seen breakthroughs [25, 3, 16, 36, 10, 11]. MONet [3] trains a Variational Auto-Encoder and a recurrent attention network to segment objects and reconstruct the image. IODINE [16] similarly relies on VAEs to infer a set of latent object representations which can be used to represent and reconstruct the scene—and like MONet uses multiple encode-decode steps. Locatello *et al.* [36] replace this procedure with a single encoding step and iterate on the attention mechanism instead, and introduce the concept of 'slot attention' to map feature representations to discrete sets of objects. All of them share the same paradigm, representing the scene as a collection of latent variables where each variable corresponds to one object. However, the decoder may produce different latent codes for objects at different locations in order to reconstruct all present objects, disregarding their possibly similar semantics, which requires higher network capacity to model complex scenes [36]. By contrast, we explicitly supervise our keypoints to form compact clusters in feature space. More recently, concurrent work [30] extended slot attention to more complex images using additional motion cues. However, it still uses synthetic datasets and cannot produce a semantically-aware latent space.

**Multiple-instance classification**. This problem differs from the above in that the objects in the scene are known a priori (*e.g.* numbers, or letters) and need not be 'discovered'. It is also related to landmark learning, but it may contain repeated objects, which breaks down the one-object-per-channel paradigm used by most methods devised for that problem. MIST [1] solves this with a differentiable top-k operation which is used to extract a predetermined number of patches, which are then fed to a task-specific network, such as a classifier. It needs to be supervised with class labels indicating how many times each instance appears, which we do not require, and unlike our approach it does not generalize to the case of an unknown or variable number of objects. We show that our method obtains similar performance to MIST and another state-of-the-art top-k selection method based on optimal transport [60].

**Unsupervised clustering**. The performance of traditional clustering methods such as k-means and Gaussian Mixture Models is highly dependent on the quality of the hand-crafted features, and it is non-trivial to incorporate them into deep networks. A popular solution is to train an autoencoder for reconstruction, and regularize the bottleneck feature to form clusters [61, 41]. These methods, however, are for the most part only applicable to image classification, because they do not have the notion of locality—*i.e.*, they produce one global description per image. Moreover, we show that naively using k-means to regulate the feature space can lead to degenerate solutions, while our carefully designed sliced Wasserstein loss does not suffer from this.

**Regularisation of Latent space**. Autoencoders [20] have been widely used for feature learning [57], but are prone to overfitting [2]. Variational autoencoders (VAE) [29] solved this problem by regularizing the learned latent space towards a certain prior distribution—normal distributions are often used. VQ-VAE [56] extends VAE to generate a discretized latent space by regularizing it with

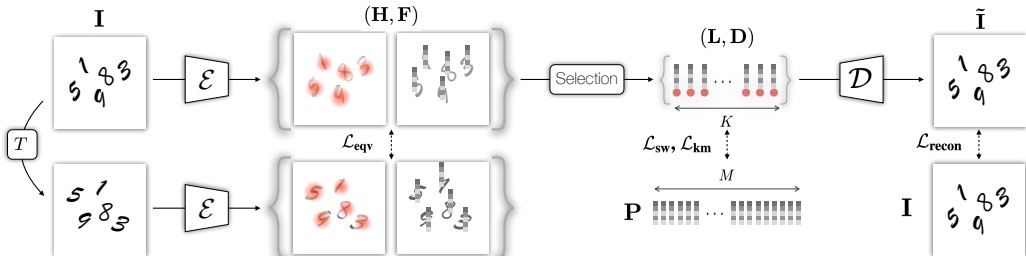

Figure 2: **Framework** – We generate a heatmap $\mathbf{H}$ and a feature map $\mathbf{F}$ through an encoder $\mathcal{E}$. We then select $K$ keypoints at locations $\mathbf{L}$ and retrieve $K$ descriptors $\mathbf{D}$ from the feature map $\mathbf{F}$, by performing non-maxima suppression on the heatmap and picking the top responses. We then decode these keypoints with the decoder $\mathcal{D}$ and reconstruct the input image. We use multiple losses for training. $\mathcal{L}_{\text{eqv}}$ enforces equivariance of heatmap and features w.r.t. a random thin plate spline transformation $T$. $\mathcal{L}_{\text{sw}}$ and $\mathcal{L}_{\text{km}}$ use learned prototypes $\mathbf{P}$ to force descriptors to form semantic clusters. $\mathcal{L}_{\text{recon}}$ minimizes the difference between image and reconstruction, similarly to an autoencoder .

a discrete codebook. Our method can also be viewed as an autoencoder, where we regularize the latent space to form semantic clusters centered around learned prototypes.

**Local features**. *Local feature* is the preferred term for 'classical' keypoint methods, which with the introduction of SIFT [38] became one of the fundamental building blocks of computer vision. They remain an open area of research [9, 55, 58, 62], but are now relegated to applications such as 3D reconstruction and re-localization [39, 52] or SLAM [42, 12]. These problems require matching arbitrary textures (corners, blobs) across a wide range of rotations and scale changes. They are thus more open-ended than the other methods presented in this section and are typically supervised from known pixel-to-pixel correspondences. Relevant to our approach are ideas such as using homographic adaptation to learn equivariant features [33, 7], or a *soft-argmax* operator [63] to select keypoints in a differentiable manner. While recent papers have drastically reduced supervision requirements [58, 55, 62], truly unsupervised learning for general-use local features remains a 'holy grail'.

## 3   Method

We illustrate our approach in Fig. 2. Given an image, we auto-encode it to obtain a latent representation of sparse keypoints. In more detail, the encoder $\mathcal{E}$ receives an input image $\mathbf{I} \in \mathbb{R}^{H \times W \times 3}$ and generates a feature map $\mathbf{F} \in \mathbb{R}^{H \times W \times C}$, preserving the spatial dimensions of the input. Unlike previous works that utilize one heatmap per keypoint, our method outputs a single heatmap $\mathbf{H} \in \mathbb{R}^{H \times W}$ on which we detect all keypoints, allowing for multiple instances of the same keypoint (or object) to be detected. From the heatmap $\mathbf{H}$, we extract $K$ keypoint locations $\mathbf{L} \in \mathbb{R}^{K \times 2}$ and their corresponding descriptors $\mathbf{D} \in \mathbb{R}^{K \times C}$ from $\mathbf{F}$. This sparse, latent representation of our autoencoder, *i.e.*, $\mathbf{L}$ and $\mathbf{D}$, is then fed into the decoder $\mathcal{D}$, which generates $\tilde{\mathbf{I}}$, a reconstruction of the input image. To force these individually detected $\mathbf{D}$ to form semantic clusters in the latent space, we learn $M$ prototypes $\mathbf{P} \in \mathbb{R}^{M \times C}$, which act as 'cluster centers' that guide the clustering process via losses to be discussed in Sec. 3.3.

This network architecture is similar to previous autoencoder-based landmark discovery papers in that we use an encoder to extract keypoint locations and descriptors and a decoder to reconstruct the image [64, 23, 37]—but our method differs in two key aspects: (a) our encoder generates a single heatmap instead of multiple ones, allowing us to operate beyond the assumption that each feature appears once and only once in every image; and (b) we regulate the latent space by clustering features around prototypes. Note that while we still need to choose a value for $K$ (and $M$), our approach can handle a variable number of objects at inference time as long as the number of extracted keypoints is sufficiently large, by filtering out low confidence detections: see CLEVR (Sec. 4.2) for results on images with a variable number of objects while using a fixed number of keypoints.

### 3.1   Encoder $\mathcal{E}$ – Keypoint and descriptor extraction

We use a U-Net-like [51] network that takes as input a unit-normalized image $\mathbf{I} \in [0, 1]^{H \times W \times 3}$ to generate a single-channel heatmap $\mathbf{H} \in \mathbb{R}^{H \times W}$ and a multi-channel feature map $\mathbf{F} \in \mathbb{R}^{H \times W \times C}$. (We found $C{=}32$ to be sufficient for all datasets and tasks.) To detect multiple instances from $\mathbf{H}$

we apply non-maximum suppression (NMS) to $\mathbf{H}$ and select the top $K$ maxima. Note that this is a non-differentiable process—inspired by LIFT [63, 45], we apply soft-argmax in the neighborhood of each maximum to obtain the final keypoint locations $\mathbf{L}$, restoring differentiability w.r.t. locations. We extract the descriptors $\mathbf{D}$ at each keypoint location from the feature map $\mathbf{F}$ via simple bilinear interpolation. Please refer to `Supplementary Material` for further details.

## 3.2 Decoder $\mathcal{D}$ – Image reconstruction from sparse features

To decode the sparse, keypoint-based representation into an image, we use another U-Net, identical to the encoder except for the number of input/output channels. We transform the sparse keypoints and descriptors into a dense representation by creating a feature map. Specifically, in order to make keypoints 'local', we generate one feature map per keypoint by copying the descriptors $\mathbf{D}$ to the locations $\mathbf{L}$ and convolving them with a Gaussian kernel with a pre-defined variance, generating $K$ feature maps. We then aggregate them with a weighted sum, according to the detection score. With this input, we train the decoder $\mathcal{D}$ to reconstruct the image $\mathbf{I}$ into $\tilde{\mathbf{I}}$.

If the object mask is necessary—*e.g.* for the object discovery task, in Sec. 4.2—instead of a weighted sum we simply reconstruct one RGB-A image for each of the $K$ feature maps and apply alpha-blending to obtain the final reconstruction $\tilde{\mathbf{I}}$. The alpha channel can be used as the object mask, required by the evaluation metrics. See `Supplementary Material` for more architectural details.

## 3.3 Learning objectives

We present various learning objectives to encourage our keypoint-based representation to have certain desirable properties. All objectives are critical to the final performance, see ablation in Sec. 4.4.

**Learning meaningful features – $\mathcal{L}_{\mathbf{recon}}$.** Similarly to a standard auto-encoder, our network learns to extract high-level features by minimizing the reconstruction loss between the input image $\mathbf{I}$ and the reconstructed image $\tilde{\mathbf{I}}$. We use either the mean squared error (MSE) or the perceptual loss [24]. We find that the perceptual loss is advantageous on real data with complex backgrounds such as CelebA [35], but the MSE loss does better on simpler datasets with no background such as CLEVR [27] or Tetrominoes [27], commonly used by object discovery papers [3, 16, 36], as well as H36M after removing the background [22].

$$\mathcal{L}_{\mathrm{recon}} = \begin{cases} \sum_l c_l \|\mathbf{V}_l(\mathbf{I}) - \mathbf{V}_l(\tilde{\mathbf{I}})\|_1, & \text{Perceptual Loss} \\ \|\mathbf{I} - \tilde{\mathbf{I}}\|_2, & \text{MSE Loss} \end{cases} \tag{1}$$

For the perceptual loss we use a pre-trained, frozen VGG16 network to extract features. $\mathbf{V}_l$ is the output of layer $l$ and $c_l$ its weighting factor, as presented in [24].

**Unsupervised clustering – $\mathcal{L}_{\mathbf{sw}}, \mathcal{L}_{\mathbf{km}}$.** In order to embed semantic meaning into our learned descriptors, we encourage descriptors to cluster in feature space. Descriptors of the same cluster should move closer together and descriptors of different clusters further apart, which is similar to the objective of contrastive learning papers [17, 5, 4]. More specifically, we learn $M$ prototypes $\mathbf{P} \in \mathbb{R}^{M \times C}$, where each prototype represents the 'center' of a class. Our goal is two-fold. First, we wish the prototypes to follow the same distribution as the features. Second, we wish the features to cluster around the prototypes. For the first objective, we minimize the sliced Wasserstein distance [6], and for the second objective, we use a k-means loss.

— *Sliced Wasserstein loss $\mathcal{L}_{sw}$.* To compute the sliced Wasserstein distance [6], we randomly sample $N$ projections $w_i$ from a unit sphere, each projecting the $C$-dimensional descriptor space into a single dimension. Then, in one-dimensional space, the Wasserstein distance $W$ can be calculated as the square distance between sorted, projected descriptors, and prototypes. The average of these $N$ distances is the sliced Wasserstein loss we seek to minimize.

The difficulty in applying this method, however, is that the number of samples from the two distributions must be equal, which is not the case for our descriptors and prototypes. We thus represent the prototypes as modes in a Gaussian Mixture Model (GMM) [8], so we can draw as many samples from the prototype distribution as the number of keypoints we extract ($K$) to compute the sliced Wasserstein loss (see `Supplementary Material` for more details). Denoting the sampled, sorted

prototypes as $\tilde{\mathbf{P}} \in \mathbb{R}^{K \times C}$ and the sorted descriptors as $\tilde{\mathbf{D}}$, we write

$$\mathcal{L}_{\text{sw}} = \frac{1}{NK} \sum_{i \in [1,N], j \in [1,K]} W(w_i^T \tilde{\mathbf{D}}_j, w_i^T \tilde{\mathbf{P}}_j). \tag{2}$$

— *k-means loss* $\mathcal{L}_{km}$. With the k-means loss, we encourage descriptors to become close to the closest prototype, relatively to the other prototypes. Inspired by [5], we minimize the log soft-min of the $\ell_2$ distances between prototypes and descriptors, where the logarithm avoids the vanishing gradients of the soft-min. We write

$$\mathcal{L}_{\text{km}} = -\frac{1}{K} \sum_{i \in [1,K]} \log \max_{j \in [1,M]} \left( \frac{\exp(-\|\mathbf{D}_i - \mathbf{P}_j\|)}{\sum\limits_{k \in [1,M]} \exp(-\|\mathbf{D}_i - \mathbf{P}_k\|)} \right). \tag{3}$$

**Making detections equivariant – $\mathcal{L}_{\text{eqv}}$.** Keypoints should be equivariant to image transformations such as affine or non-rigid deformations. This idea appears in several local feature papers [7, 48], and unsupervised landmark learning papers [53, 64, 23]. Following [28, 53] we apply a thin plate spline transformation $T$ and use the $l_2$ distance as a loss to ensure that both heatmap and feature map are equivariant under non-rigid deformations. Note that the encoder $\mathcal{E}$ produces a heatmap and a feature map and we apply the loss to both separately—we use a single equation for simplicity:

$$\mathcal{L}_{\text{eqv}} = \|\mathcal{E}(T(\mathbf{I})) - T(\mathcal{E}(\mathbf{I}))\|_2. \tag{4}$$

**Iterative training**. We found that straightforward training of our framework leads to convergence difficulties. We thus alternate training of descriptors and prototypes. Specifically, we train only the encoder and the decoder for one step via $\lambda_{\text{recon}} \mathcal{L}_{\text{recon}} + \lambda_{\text{km}} \mathcal{L}_{\text{km}} + \lambda_{\text{eqv}} \mathcal{L}_{\text{eqv}}$ (see `Supplementary Material` for details). We then train the prototypes and their GMM for eight steps using $\mathcal{L}_{\text{sw}}$. We repeat this iterative training process until convergence.

## 4  Results

We apply our method to three tasks: multiple-instance object detection, object discovery, and landmark detection. We use five different datasets.

**MNIST-Hard [1]**  contains synthetically-generated images composed of multiple MNIST digits. We follow the procedure in [1] and put 9 MNIST digits on a black canvas at random positions, sampling from a Poisson distribution. Variations of the same digit may co-occur in the same image. We generate 50K such images for training and testing, respectively.

**CLEVR[27]**  contains visual scenes with a variable number of objects in each scene. Following [16, 3, 36], we only use the first 70K images that contain at most 6 objects (also known as CLEVR6). We train our model with the first 60K images and evaluate with the last 10K.

**Tetrominoes[27]**  contains 80K images of Tetrominoes, a geometric shape composed of four squares. Each image contains three randomly sampled, non-overlapping Tetrominoes. There are 19 unique tetrominoes shapes (including rotations). Each Tetromino can be in one of the 6 possible colors. We train our model using the first 50K images and evaluate with the last 10K.

**CelebA [35]**  contains 200k images of human faces. Earlier efforts [37] used aligned images (with centered faces) for both training and evaluation, a dataset that is now heavily saturated. We follow [34] instead, which uses CelebA with unaligned images, filtering out those where the face accounts for less than 30% of the pixels. We follow the standard procedure and train our model without supervision using all images except for the ones in the MAFL (Multi-Attribute Facial Landmark) test set, and train a linear regressor which regresses the 5 manually annotated landmarks, on the MAFL training set. We report pixel accuracy normalized by inter-ocular distance on the MAFL test set.

**Human 3.6M (H36M) [22]**  contains 3.6M captured human images with ground truth joint locations from 11 actors (7 for training, 4 for test) and 17 activities. We follow [64], using 6 actors in the training set for training and the last one for evaluation. We subtract the background with a method provided by the dataset. As for CelebA, we first train our model on the training set and then learn a linear regressor that predicts the location of 36 joints. Since the front and back views are often similar, we follow [64] and swap left and right joint labels on images where the subject is turned around.

Table 1: **Object detection on MNIST-Hard.** We compare our method against weakly supervised methods (class labels are available during training) and fully supervised methods (location and class label are available during training). Our method performs best in terms of localization and similarly to Xie *et al.* [60], although our approach is fully unsupervised.

| | *Unsupervised* | *Weakly Supervised* | | | *Fully Supervised* |
|---|---|---|---|---|---|
| | Ours | MIST[1] | Xie *et al.* [60] | Ch.-wise | CNN |
| Localization | 99.9% | 97.8% | 72.7% | 25.4% | 99.6% |
| Classification | 92.1% | 98.8% | 93.1% | 75.5% | 98.7% |
| Both (Loc. ∩ Classif.) | 92.1% | 97.5% | 71.3% | 24.8% | 98.6% |

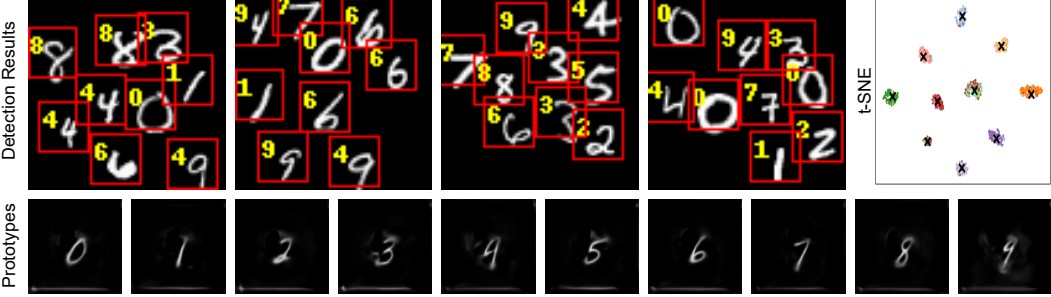

Figure 3: **Qualitative results on MNIST-Hard. (Top left)** Detection results on MNIST-Hard dataset. Our method also tends to misclassify digits when two different digits are similar (4 is misclassified to 9 on the second image). **(Top right)** the t-SNE plot of latent space. The coloured ● are extracted latent vectors where colour indicates the ground truth label for each digit, while black **X** are learned prototypes. **(Bottom)** Visualizations of learned prototypes. Note that prototypes for 4 and 9 are similar, sometimes causing misclassifications on these digits.

## 4.1 Multiple-instance classification and detection: MNIST-Hard [1]

MNIST-Hard contains nine instances of ten different digits—we thus extract $K=9$ keypoints and $M=10$ prototypes to represent them. The main challenge for this task is that the images may contain multiple instances of the same digit. Note that all the baseline methods used in the landmark detection task (Sec. 4.3) follow the one-class-per-channel paradigm and will thus fail in this scenario. This limitation is also mentioned in MIST [1], which reports that a channel-wise approach, using the same auto-encoder as MIST while replacing their single-heatmap approach with $K$ separately-predicted channels, results in poor performance: we list this baseline as 'Ch.-wise'. We also consider [60], a differentiable top-k operator based on optimal transport, and a fully supervised version of MIST with both location and class annotation available during training that acts as a performance upper bound.

To evaluate detection accuracy we compare the ground-truth bounding boxes to our predictions by placing bounding boxes the approximate size of a digit ($20 \times 20$px) around each keypoint. We consider a prediction to be correct with an intersection over union (IoU) of at least 50%. For the classification task, we assign prototypes to each class through bipartite matching, while maximizing accuracy. We also report the intersection of the two metrics in Tab. 1. Our method outperforms all baselines on the detection task and is close to MIST on the classification task. Note that ours is the only unsupervised method: all baselines have access to class labels, or class labels and location. Qualitative results for detections, prototypes, and t-SNE [40] visualizations are shown in Fig. 3.

## 4.2 Object Discovery: CLEVR [27] and Tetrominoes [27]

We also consider a multi-object discovery task using two different datasets: CLEVR and Tetrominoes. Performance is evaluated with the Adjusted Rand Index (ARI) [47, 21], which computes the similarity between two clusterings, adjusted for the chance grouping of elements, ranging from 0 (random) to 1 (perfect match). This requires semantic masks, so for this task, we modify our network as outlined in Sec. 3.2. Our masks are generated by applying an argmax operator over the alpha channels of the $K$ per-keypoint reconstructions, and are subsequently used to compute ARI. In addition to the standard metric, we report the classification performance of our method, based on shape, color, and size (the latter only for CLEVR)—which none of the baselines are capable of. To calculate classification

Table 2: **Quantitative results for CLEVR and Tetrominoes.** Adjusted Rand Index (ARI) scores for CLEVR and Tetrominoes (mean ± standard deviation, across five different seeds). We do not report standard deviation for our method, which is deterministic at inference. Our approach is on par with the state of the art in ARI, and can additionally perform unsupervised classification. Note that for CLEVR, shape classification performance is lower because the dataset contains 3D motion (near/far), and some prototype capacity is used to explain this rather than shape, as more pixels change by moving an object from the near to the back plane than *e.g.* moving between 'sphere' and 'cube'. This lack of 3D spatial information is currently a limitation of our method.

| | CLEVR6 | | | | Tetrominoes | | |
| | ARI | Classification accuracy | | | ARI | Classification accuracy | |
| | | Shape | Color | Size | | Shape | Color |
|---|---|---|---|---|---|---|---|
| IODINE [16] | 98.8 ± 0.0 | - | - | - | 99.2 ± 0.4 | - | - |
| MONet [3] | 96.2 ± 0.6 | - | - | - | - | - | - |
| Slot Attention [36] | 98.8 ± 0.3 | - | - | - | 99.5 ± 0.2 | - | - |
| Ours | 98.3 | 46.8 | 90.7 | 97.1 | 99.7 | 91.3 | 100 |

Figure 4: **Visualization on CLEVR.** Because our method does not explicitly model the background, the semantic masks have vague boundaries. However, the individual alpha channel shows accurate object masks. **(Top)** Our method successfully captures all six objects in the image. Notice that the green cylinder on the left is partially occluded by the red sphere, but as we enforce features to be close to prototypes, the individually rendered images show a full cylinder, and the occlusion information is embedded in the alpha map. **(Bottom)** Four objects are detected with high confidence and the remaining two are unused, showing that how our approach can handle a variable number of objects.

accuracy, as our method is fully unsupervised, we simply assign, for each prototype, the properties (shape, color, size) that best represent the prototype. We use $K=6$ keypoints and $M=48$ prototypes for CLEVR and $K=3$ and $M=114$ for Tetrominoes, which are the maximum number of objects and the number of combinations of shape, color, and size, respectively. We report the results in Table 2, showing that our method performs on par with the state of the art, and additionally performs unsupervised classification. We show qualitative examples and segmentation masks in Figs. 4 and 5.

## 4.3 Landmark Detection: CelebA-in-the-wild [35] and H36M [22]

We extract $K=4$ keypoints with $M=32$ prototypes for CelebA and $K=16$ keypoints with $M=32$ prototypes for H36M. As noted previously, we follow [53, 64, 23, 37] for evaluation and train a linear layer without bias term using ground truth landmark annotations to map the learned landmarks to a set of human-defined landmarks, so that the estimated landmarks are entirely dependent on the detected keypoints. We place the frame of reference for the keypoints at the top left corner of the image to be consistent with previous papers. In existing land-

Table 3: Performance on CelebA-in-the-wild and Human3.6M. The error is normalized by interocular distance for CelebA and by image size for H36M.

| | | CelebA ($K=4$) | H36M ($K=16$) |
|---|---|---|---|
| | Thewlis [53] | - | 7.51% |
| | Zhang [64] | - | 4.91% |
| Single obj. | Jakab [23] | 19.42% | - |
| | Lorenz [37] | 15.49% | 2.79% |
| | He [19] | 25.81% | - |
| | He [19] (tuned) | 12.10% | - |
| Multi obj. | Ours | 18.49% | 6.88% |

mark detection methods, the linear layer has an input size $K \times 2$ and an output size $T \times 2$, where $K$ is the number of heatmap channels and $T$ is the number of human-defined landmarks, which is possible, as they assume a keypoint to be detected precisely once. This is not directly applicable to our method, as we may detect more than one keypoint belonging to the same prototype—*e.g.*, in an extreme case, all $K$ keypoints could belong to a single prototype.

To support multiple detections, we form the input to our linear layer as a $M \times K \times 2$ tensor, which is populated while preserving prototype information through the indexing: for instance, if $n$ keypoints

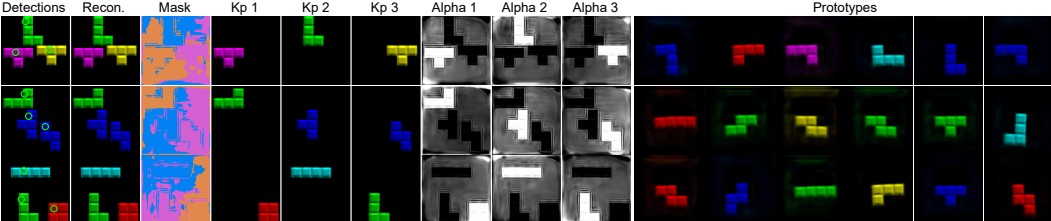

Figure 5: **Tetrominoes dataset.** Each prototype represents unique shape-color combination.

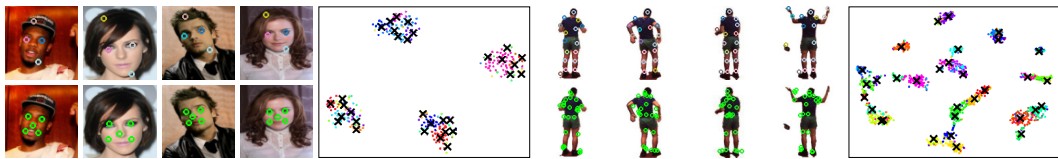

Figure 6: **Results on CelebA and H36M.** We show detection results on columns one and three, with the top row displaying the points detected by our approach, and the bottom row those generated by the regressor. Columns two and four show the clustering of prototypes and features using t-SNE.

belong to prototype $k$, we first sort them by their x coordinate from left to right, and then fill the sorted keypoint coordinates into the input tensor at $[k, 0]$ to $[k, n-1]$. Unused entries are simply set to zero—note that only $K$ coordinates are non-zero, the same as for baseline methods.

We report quantitative results in Tab. 3 and show detections and clustering results in Fig. 6. Our method provides comparable performance to the methods targeting single objects but is still able to deal with multiple instances, which none of the baseline methods can do; see Fig. 1. While there is still a gap between our method and the best performing single-instance landmark detector, which is a limitation of our method, considering how the single-instance assumption simplifies the problem, our results are promising.

## 4.4 Ablation study

### 4.4.1 Effect of different loss terms

We rely on multiple loss terms and iterative training. Tab. 4 reports their effect on MNIST-Hard (Sec. 4.1), and

Table 4: Ablation on the loss function and training strategy.

| $\mathcal{L}_{\text{recon}}$ | $\mathcal{L}_{\text{sw}}$ | $\mathcal{L}_{\text{km}}$ | $\mathcal{L}_{\text{eqv}}$ | Iterative training | Classif. | Localization | Both |
|---|---|---|---|---|---|---|---|
| ✓ | | | | ✓ | 10.3% | 99.9% | 10.3% |
| ✓ | ✓ | | | ✓ | 40.7% | 99.9% | 40.7% |
| ✓ | ✓ | ✓ | | ✓ | 78.4% | 99.6% | 78.0% |
| ✓ | ✓ | ✓ | ✓ | | 71.6% | 99.9% | 71.6% |
| ✓ | ✓ | ✓ | ✓ | ✓ | 92.8% | 99.9% | 92.8% |

Fig. 7 shows the corresponding results in the quality of the clustering with t-SNE [40]. Note that the reconstruction loss is used in all the runs in Tab. 4, as it is the main loss term used to train the encoder and decoder—our method will not work without it. The other loss terms are used to regulate the latent space and encourage clustering, which has a significant impact on learning semantically meaningful descriptors.

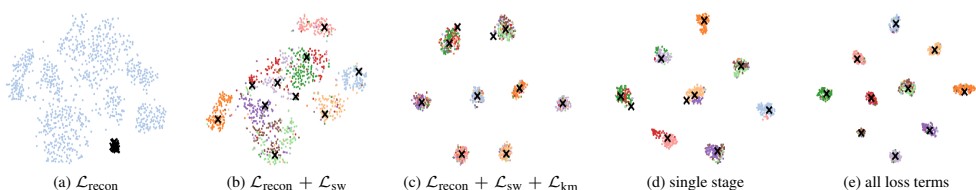

(a) $\mathcal{L}_{\text{recon}}$     (b) $\mathcal{L}_{\text{recon}} + \mathcal{L}_{\text{sw}}$     (c) $\mathcal{L}_{\text{recon}} + \mathcal{L}_{\text{sw}} + \mathcal{L}_{\text{km}}$     (d) single stage     (e) all loss terms

Figure 7: **Ablation study w.r.t. clustering results.** We illustrate the effect of the different components used for training by showing the clustering of features and prototypes with t-SNE. Features are indicated by ● (color indicates ground truth label), and prototypes as ✕. (a) vs (b) shows that the sliced Wasserstein loss is necessary to move the prototypes so they match the distribution of the features; (b) vs (c) shows that the clustering loss encourages the feature space to form clusters; (d) vs (e) shows the importance of our iterative training approach; (c) vs (e) shows that the equivariance loss further improves the result. Note that these five plots correspond to the five rows in Tab. 4.

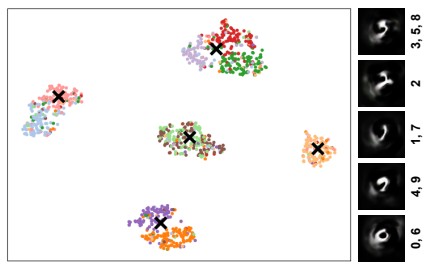 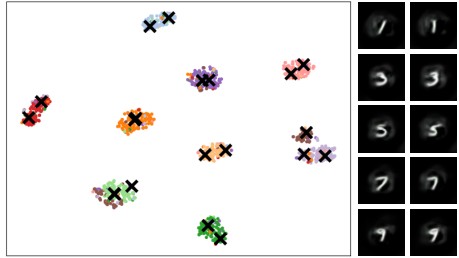

(a) **5 prototypes:** Each prototype learns to represent multiple digits.

(b) **20 prototypes:** Prototypes learn to represent different modes for the digits (shown: five).

Figure 8: **Different number of prototypes on MNIST-Hard. (Left)** Features are indicated by • (color indicates ground truth label), prototypes as ×. **(Right)** Visualizations of the prototypes.

Table 5: **Ablation on number of prototypes**

|  | $M$=5 | $M$=5 (reverse) | $M$=10 | $M$=20 |
|---|---|---|---|---|
| Localization | 90.8% | 90.8% | 99.9% | 99.9% |
| Classification | 47.4% | 98.3% | 92.8% | 87.7% |
| Both | 43.0% | 89.2% | 92.8% | 87.7% |

Table 6: **Ablation on clustering strategy**

|  | Classification | Localization | Both |
|---|---|---|---|
| K-means | 21.1% | 99.5% | 99.9% |
| Vector Quantization | 23.7% | 65.3% | 15.3% |
| Sliced Wasserstein | 92.8% | 99.9% | 92.8% |

### 4.4.2 Number of Prototypes

We ablate the number of prototypes on MNIST-Hard in Tab. 5. If it is smaller than the number of classes, each prototype learns to represent multiple classes, as shown in Fig. 8(a). With 5 prototypes representing 10 classes, classification accuracy will be at most 50%: we achieve 47.4%. We can also evaluate the reverse mapping from class label to prototypes and check the consistency of assignments, which gives us 98.3% (note that this is not accuracy). With more prototypes than classes, prototypes learn to represent different modes of the class, as shown in Fig. 8(b). In general, a reasonable result can be achieved if the number of prototypes $M$ is larger than the number of classes.

### 4.4.3 Clustering Strategy

We investigate the importance of the sliced Wasserstein loss in Tab. 6 by replacing it with a K-means loss used in [41] and a quantization loss used in [56], while keeping everything else the same. The K-means loss does not encourage a balanced assignment among features and prototypes, and all the features may collapse to a single prototype, resulting in a low classification score. The Vector Quantization loss discretizes the latent space using a code-book—said codes are similar to our learned prototypes. However by enforcing same-class objects to be encoded into a single code, the network loses the ability to model intra-class variance. By contrast, our method uses prototypes to model inter-class variance, modeled with a Gaussian distribution.

## 5 Conclusion

We introduce a novel method able to learn keypoints without any supervision signal. We improve on prior work by removing the key limitation of predicting separate channels for each keypoint, which requires isolated instances . We achieve this with a novel feature clustering strategy based on prototypes, and show that our method can be applied to a wide range of tasks and datasets. We provide a discussion on the potential societal impact of our method in the `Supplementary Material`.

**Limitations**. Our approach does not have a concept of 3D structure and has limited applicability to scenarios where scene understanding requires compositionality (see shape classification on Tab. 2). It is also unable to model the background. We plan to address these limitations in future work.

## Acknowledgments and Disclosure of Funding

This work was supported by the Natural Sciences and Engineering Research Council of Canada (NSERC) Discovery Grant / Collaborative Research and Development Grant, Google, Digital Research Alliance of Canada, and Advanced Research Computing at the University of British Columbia.

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
