# A Supplementary material

The appendix includes a detailed description of the soft-argmax used in encoder $\mathcal{E}$ in Sec. A.1, the sampling strategy for prototypes and pseudo-code for prototype training in Sec. A.2, model generalization on CLEVR10 in Sec. A.3, architecture details in Sec. A.4, a table of all the hyperparameters for the different datasets in Sec. A.5, more qualitative results on occlusion modelling in Sec. A.6, a summary of the computational resources used to train the models in Sec. A.7, dataset licenses in Sec. A.8, and a discussion on potential societal impact in Sec. A.9.

## A.1 Detailed Description of the Soft-Argmax in Encoder $\mathcal{E}$

In the encoder $\mathcal{E}$, we first generate a scoremap $\mathbf{S}$ the same shape as input image $\mathbf{I}$ with a U-Net-style network, followed by a sigmoid activation, whose output is in $[0, 1]$. Then, we apply NMS on the scoremap $S$ and select top $K$ keypoints $\mathbf{T}$ from it. However, this top-k selection is non-differentiable—specifically, the gradients only flow back through the $K$ locations on the $H \times W$ scoremap $S$. To avoid sparsity in the gradient, we construct a kernel of size $B \times B$ centered at each keypoint location and use a soft-argmax [63] function to compute the final keypoint location as a weighted sum of the region around the center of the kernel. By using these weighted centers, the gradient flows back through $B \times B$ locations for each keypoint, rather than just one. As a by-product, we also obtain sub-pixel accuracy on the keypoints.

$$\mathbf{L_i} = \frac{\sum_{\mathbf{c} \in kernel\{\mathbf{T_i}\}} \mathbf{c} \cdot \exp\left(\mathbf{S}[\mathbf{c}^x, \mathbf{c}^y]/\tau\right)}{\sum_{\mathbf{c} \in kernel\{\mathbf{T_i}\}} \exp\left(\mathbf{S}[\mathbf{c}^x, \mathbf{c}^y]/\tau\right)} \tag{5}$$

where $kernel\{\mathbf{T_i}\}$ are the points within a kernel of size $B \times B$ centered at point $\mathbf{T_i}$, and $\tau$ is a hyper parameter to control the hardness of the softmax operation.

## A.2 Prototype Learning

### A.2.1 Sampling for Sliced Wasserstein Loss

In order to apply the sliced Wasserstein loss to match the distribution of the prototypes to the distribution of the descriptors, we require an equal number of prototypes and descriptors. Thus, we model the descriptors as a Gaussian Mixture Model (GMM) with a small predefined variance $\Sigma$ (a hyperparameter), since we would ideally want the descriptors to form very sharp modes around the Gaussian centers. Thus, denoting the sampled descriptors from the prototype GMM as $\tilde{\mathbf{D}}$ we sample $B \times K$ samples from

$$p(\tilde{\mathbf{D}}_i) = \sum_{j \in [1, M]} \pi_j \mathcal{N}(\tilde{\mathbf{D}}_i | \mathbf{P}_j, \Sigma), \tag{6}$$

where $\pi_j$ are the mixture weights of the GMM for the $j$-th prototype. For the mixture weight $\pi_j$, note that each prototype may have a different number of descriptors associated with it. We thus define it according to the ratio of descriptors that are associated with each prototype—we associate via finding the nearest prototype with the $\ell_2$ norm—but with a term that encourages exploration when the compactness of the prototypes is not equal. For example, when a certain prototype dominates but is widely spread, we would ideally want to explore using not just this single prototype. Mathematically, denoting the ratio of descriptors associated with prototype $m$ as $r_m$, and the variance of the descriptors associated with the prototype as $\sigma_m$, we write

$$\alpha_m = r_m + Var(\{\sigma_i, i \in [1, M]\}), \tag{7}$$

and

$$\pi_m = \frac{\alpha_m}{\sum_{m \in [1, M]} \alpha_m}. \tag{8}$$

For prototypes without any descriptors assigned, we simply set $\sigma_m = 1$.

### A.2.2 Pseudo-code for the prototype learning algorithm

As stated in Sec. 3.3, within each training iteration, we first optimize encoder and decoder with fixed prototypes. We then optimize the prototypes with a fixed encoder/decoder. Here we provide a detailed pseudo-code version of the algorithm we use for prototype optimization with a sliced Wasserstein loss, shown in Tab. 7.

Table 7: **Prototype Training:** Pseudo code for prototype training.

---

**Requirement:** $\mathbf{D}$: descriptors, $M$: number of descriptors, $\mathbf{P}$: prototypes,
$N$: number of prototypes.

---

**Function** TrainPrototype $(\mathbf{D}, \mathbf{P})$
1. Calculate mixing ratio $\pi_m$
   - $\mathbf{D}_m \leftarrow$ divide $\mathbf{D}$ into $M$ subsets where each subset is associated to a member in $\mathbf{P}$ in terms of smallest $\ell_2$ norm
   - $r_m \leftarrow$ calculate the ratio of $\mathbf{D}_m$ in $\mathbf{D}$
   - $\sigma_m \leftarrow$ calculate variance of $\mathbf{D}_m$
   - $\sigma \leftarrow$ calculate variance of $\sigma_m$
   - $\alpha_m \leftarrow r_m + \sigma$
   - $\pi_m \leftarrow \alpha_m / \sum \alpha_m$
2. Sample from GMM
   - $\tilde{\mathbf{P}} \leftarrow$ initiate empty list
   - **for** $p_m$ **in** $\mathbf{P}$ :
   -    append $\pi_m \times N$ samples from a Gaussian centered at $p_m$ with a predefined variance to $\tilde{\mathbf{P}}$
3. Calculate sliced Wasserstein distance
   - $d \leftarrow SW_{distance}(\tilde{\mathbf{P}}, \mathbf{D})$
4. Train prototypes
   - Optimize $\mathbf{P}$ by minimizing $d$

---

Table 8: **Quantitative results for CLEVR6 and CLEVR10.** We train the model on CLEVR6, and evaluate on both CLEVR6 and CLEVR10.

|          | ARI  | Classification accuracy | | |
|----------|------|-------|-------|------|
|          |      | Shape | Color | Size |
| CLEVR6   | 98.6 | 53.5  | 91.0  | 95.8 |
| CLEVR10  | 97.6 | 52.5  | 90.7  | 97.9 |

## A.3 Generalization to CLEVR10

We demonstrate that our approach generalizes to a larger number of instances without any retraining in Tab. 8, where we follow [36] to evaluate our CLEVR6 -trained model on CLEVR10 (which contains up to 10 objects). Our model can deal with a larger number of instances and generalizes very well, with only a very small drop in performance. We do not report numbers for the baselines as they are not available.

## A.4 Architecture Detail

We summarize all the main components in our pipeline in Tab. 9.

## A.5 Hyperparameter Table

We report the hyperparameters we use for each dataset/task in Tab. 10. The number of prototypes and keypoints are set according to the dataset characteristics. The softmax kernel size was also adjusted to match the dataset image size and the rough size of the object of interest. Batch sizes were determined according to the memory limit of our GPU. Except for CelebA, all settings are similar

.

Table 9: **Architecture Detail.** A list of all the main components in our pipeline.

| General Architecture | Architecture for Object Discovery task |
|---|---|
| **Encoder $\mathcal{E}$** | |
| $\mathbf{I} \in [0,1]^{H \times W \times 3}$ | |
| $UNet(\mathbf{I}) \to \mathbf{H} \in \mathbb{R}^{H \times W}, \mathbf{F} \in \mathbb{R}^{H \times W \times 32}$ | |
| $Sigmoid(\mathbf{H}) \to \mathbf{S} \in [0,1]^{H \times W}$ | |
| **Keypoints Sampling** | |
| $NMS(\mathbf{S}) \to \tilde{\mathbf{S}} \in [0,1]^{H \times W}$ | |
| $Top\text{-}K(\tilde{\mathbf{S}}) \to \mathbf{P} \in \mathbb{N}^{K \times 2}$ | |
| $Soft\text{-}Argmax(\mathbf{P}) \to \mathbf{L} \in \mathbb{R}^{K \times 2}$ | |
| $Bilinear\ Sampling(\mathbf{F}, \mathbf{L}) \to \mathbf{D} \in \mathbb{R}^{K \times 32}$ | |
| **Sparse Reconstruction** | |
| $Gaussian\ Conv.(\mathbf{L}, \mathbf{D}) \to \mathbf{R} \in \mathbb{R}^{K \times H \times W \times 32}$ | |
| $Summation(\mathbf{R}) \to \tilde{\mathbf{F}} \in \mathbb{R}^{H \times W \times 32}$ | - |
| **Decoder $\mathcal{D}$** | |
| $UNet(\tilde{\mathbf{F}}) \to \tilde{\mathbf{I}} \in [0,1]^{H \times W \times 3}$ | $UNet(\mathbf{R}) \to \mathbf{A} \in [0,1]^{H \times W \times 4}$ |
| - | $Alpha\text{-}Blending(\mathbf{A}) \to \tilde{\mathbf{I}} \in [0,1]^{H \times W \times 3}$ |

Table 10: **List of Hyper Parameters.**

| | MNIST | CLEVR6 | Tetrominoes | CelebA | H36M |
|---|---|---|---|---|---|
| # Keypoints | 9 | 6 | 3 | 4 | 16 |
| # Prototypes | 10 | 48 | 114 | 32 | 32 |
| Softmax Kernel Size | 13 | 21 | 27 | 21 | 11 |
| $\Sigma$ in GMM | | | 4e-4 (all) | | |
| Recon. Loss Type | MSE | MSE | MSE | Perceptual | MSE |
| Coef. Recon. Loss | | | 1 (all) | | |
| Coef. Cluster Loss | | | 0.01 (all) | | |
| Coef. Eqv. Heatmap Loss | 0.01 | 0.01 | 0.01 | 0.05 | 0.01 |
| Coef. Eqv. Featuremap Loss | 100 | 100 | 100 | 500 | 100 |
| Encoder & Decoder Learning Rate | | | 0.001 (all) | | |
| Prototype Learning Rate | | | 0.1 (all) | | |
| Batch Size | 40 | 64 | 76 | 32 | 32 |

## A.6 Additional results on CLEVR dataset with occluded

In Fig. 4 we mentioned that our model can embed the occlusion information in the alpha channel. Unfortunately, the CLEVR dataset does not provide any annotation that can be used to evaluate occlusion quantitatively. Instead, we include more qualitative results in Fig. 9.

## A.7 Computation Resources

We train all our models on NVIDIA V100 GPUs with 32GB of RAM. Training on MNIST converges within 12 hours on a single GPU. For the other four datasets, we use two GPUs, which allows for larger batch size, for about 24 hours.

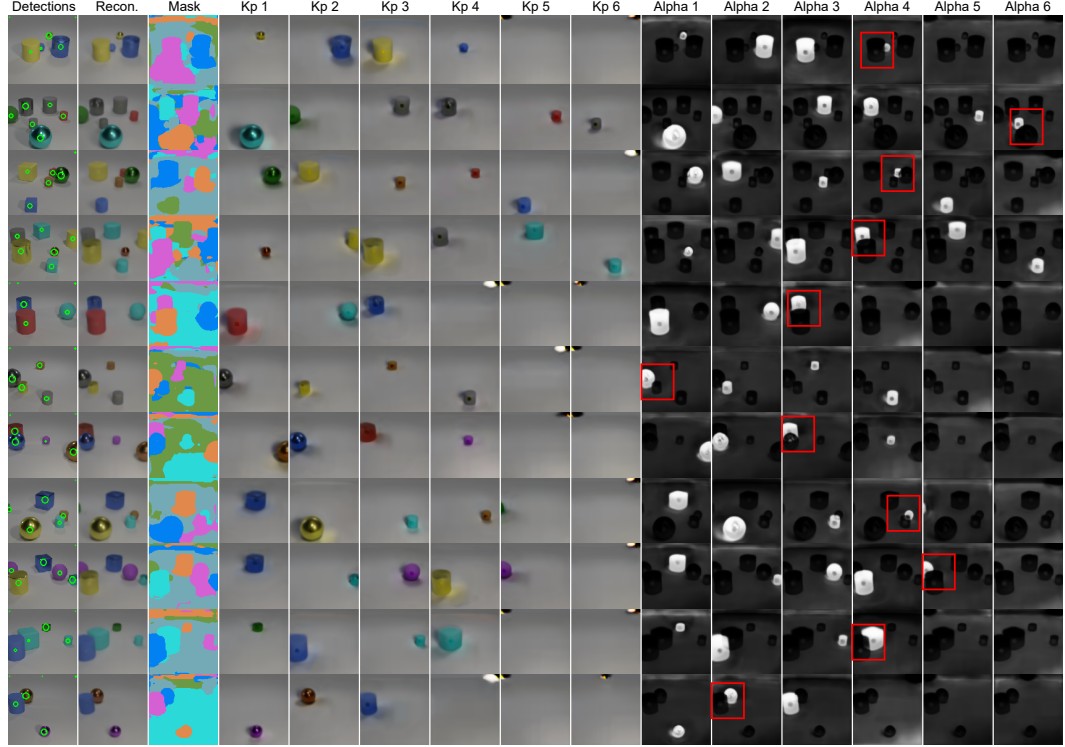

Figure 9: **More qualitative results on modelling occlusion.** Occluded regions are marked with red bounding boxes.

## A.8 Dataset Licenses

### A.8.1 MNIST http://yann.lecun.com/exdb/mnist/

The MNIST-Hard dataset is derived from the MNIST dataset. Below is the license for the original MNIST dataset:

Yann LeCun and Corinna Cortes hold the copyright of MNIST dataset, which is a derivative work from the original NIST datasets. The MNIST dataset is made available under the terms of the Creative Commons Attribution-Share Alike 3.0 license.

### A.8.2 CLEVR[27] https://github.com/deepmind/multi_object_datasets

Apache-2.0 license.

### A.8.3 Tetrominoes[27] https://github.com/deepmind/multi_object_datasets

Apache-2.0 license.

### A.8.4 CelebA[35] http://mmlab.ie.cuhk.edu.hk/projects/CelebA.html

1. CelebA dataset is available for non-commercial research purposes only.
2. All images of the CelebA dataset are obtained from the Internet which are not property of MMLAB, The Chinese University of Hong Kong. The MMLAB is not responsible for the content nor the meaning of these images.
3. You agree not to reproduce, duplicate, copy, sell, trade, resell or exploit for any commercial purposes, any portion of the images and any portion of derived data.
4. You agree not to further copy, publish or distribute any portion of the CelebA dataset. Except, for internal use at a single site within the same organization it is allowed to make copies of the dataset.

5. The MMLAB reserves the right to terminate your access to the CelebA dataset at any time.

6. The face identities are released upon request for research purposes only. Please contact us for details.

### A.8.5 H36M[22] http://vision.imar.ro/human3.6m/description.php

1. GRANT OF LICENSE FREE OF CHARGE FOR ACADEMIC USE ONLY Licenses free of charge are limited to academic use only. Provided you send the request from an academic address, you are granted a limited, non-exclusive, non-assignable and non-transferable license to use this dataset subject to the terms below. This license is not a sale of any or all of the owner's rights. This product may only be used by you, and you may not rent, lease, lend, sub-license or transfer the dataset or any of your rights under this agreement to anyone else.

2. NO WARRANTIES The authors do not warrant the quality, accuracy, or completeness of any information, data or software provided. Such data and software is provided "AS IS" without warranty or condition of any nature. The authors disclaim all other warranties, expressed or implied, including but not limited to implied warranties of merchantability and fitness for a particular purpose, with respect to the data and any accompanying materials.

3. RESTRICTION AND LIMITATION OF LIABILITY In no event shall the authors be liable for any other damages whatsoever arising out of the use of, or inability to use this dataset and its associated software, even if the authors have been advised of the possibility of such damages.

4. RESPONSIBLE USE It is YOUR RESPONSIBILITY to ensure that your use of this product complies with these terms and to seek prior written permission from the authors and pay any additional fees or royalties, as may be required, for any uses not permitted or not specified in this agreement.

5. ACCEPTANCE OF THIS AGREEMENT Any use whatsoever of this dataset and its associated software shall constitute your acceptance of the terms of this agreement. By using the dataset and its associated software, you agree to cite the papers of the authors, in any of your publications by you and your collaborators that make any use of the dataset, in the following format (NOTICE THAT CITING THE DATASET URL INSTEAD OF THE PUBLICA-TIONS, WOULD NOT BE COMPLIANT WITH THIS LICENSE AGREEMENT):

### A.8.6 Multi-face images

All multi-face images are in the public domain.

### A.9 Societal Impact

Our method would be a front-end for a typical computer vision pipeline and thus is not immediately linked to any particular application with significant societal impact. Nonetheless, our method would facilitate unsupervised learning from images which could have significant impact down the road. Unsupervised learning would significantly reduce the need for human effort within any automated workflow, which would bring both convenience and a certain amount of change—the latter may require careful consideration when adopting these technologies more widely.