# OpenReview forum: "TUSK: Task-Agnostic Unsupervised Keypoints"
_NeurIPS.cc/2022/Conference — NeurIPS 2022 Accept_

### Official Review · Reviewer_q83N · 2022-06-29

**Rating:** 7
**Confidence:** 4
**Soundness:** 3 good
**Presentation:** 3 good
**Contribution:** 2 fair

**Summary:**

The paper describes a method for learning keypoint detectors in an unsupervised manner. The key extension over related work is that the proposed method does not build on the assumption that only one target keypoint is present in an image. Technically this is achieved by having one keypoint heatmap/response map in the architecture that is shared among all keypoints, whereas related work estimates one heatmap per keypoint. Other technical strategies that are used to encourage equivariance and to clustering of the feature descriptors have also been used in prior work, but are combined in this work to enable fully unsupervised learning of keypoints detectors. The experimental evaluation shows that the proposed method performs comparably to related work on rather artificial datasets and simple datasets of faces and humans, while not building on the single object assumption.

**Questions:**

I would like the authors to comment on the weaknesses that I described. I will carefully reconsider my rating depending the authors response especially concerning the lack of novelty and simplicity of the datasets in the experiments, as well as the discussion with the other reviewers.

**Limitations:**

The limitation section addresses the core limitations adequately and concisely.

**Strengths And Weaknesses:**

Strengths:
+ The paper addresses an important problem, in that it aims to resolve the major issue that so far methods for unsupervised keypoint discovery assume that only one object is present in an image. While this is a valid assumption when learning from video streams (where objects can be segmented fairly well based on their movement), it is a very limiting factor when learning from unordered sets of images. I have myself been concerned about this issue for some time, and I appreciate that this paper makes progress towards resolving this limitation.
+ The paper is very well written and describes the key concepts very intuitively. I have enjoyed reading the paper and could understand the key concepts already after the first reading pass.

Weaknesses:
- Key concepts presented and combined in this paper are very well known. While using a single heatmap to represent the activation of all keypoints to resolve the assumption about keypoint numbers is relevant, the contrastive learning of feature representations, clustering of feature representations, and equivariance losses are well known and widely applied.
- The experimental evaluation is limited as it is conducted on very simple datasets. I understand that this is meant to be a proof-of-concept evaluation, but given the limited variability of the object shapes, appearance and background it remains unclear if much more simple method  simple clustering of features [C,E] would suffice. I highly recommend testing the proposed method on datasets of real-world images with more complex variations in shape and appearance  such as PASCAL3D+ or VehiclePart [D] or even KITTI.
- Several related works that for unsupervised learning of landmark and part representations are not mentioned [A-C] and should be compared to.
- The claim that the proposed method can model occlusion (e.g. l330) is not quantified and only shown qualitatively on images with very minor occlusion. I would recommend to tone down this claim as it seems unjustified.

[A] Unsupervised learning of object landmarks by factorized spatial embeddings.

[B] Unsupervised learning of object frames by dense equivariant image labelling.

[C] Visual concepts and compositional voting

[D] Detecting semantic parts on partially occluded objects

[E] Learning deep parsimonious representations

=====================POST REBUTTAL=========================
After reading the other reviews and the rebuttal I vote for accepting the paper. My concerns have been addressed by the additional results and clarifications provided in the rebuttal. I also think that the concerns of the other reviewers were addressed sufficiently. Therefore I raise my initial score to 7.

---

### Official Review · Reviewer_9vPX · 2022-07-07

**Rating:** 4
**Confidence:** 4
**Soundness:** 2 fair
**Presentation:** 2 fair
**Contribution:** 3 good

**Summary:**

This paper proposed a new method for learning task-agnostic keypoints in an unsupervised way. The proposed method encodes semantics into the keypoints by reconstructing the original image from sparse keypoints descriptors. A group of prototypes is learned during training and the descriptors are constrained to be around the prototypes. The proposed method can predict keypoints for multiple instances and achieves comparable results with the state of the art.

**Questions:**

1. In Table 4, what will happen if the L_recon or L_sw is removed from the entire loss?
2. In Table4, the localization results seem to change slightly with different settings of losses. Is it because the localization is too easy on the MNIST-Hard dataset.
3. Some typos need to be refined, e.g.
L. 67, 'priori'->'prior'
In the supplementary L. 532, P_i represents both point position and prototype.
In the supplementary L. 537, should 'descriptors' be 'prototype'?

**Limitations:**

The authors have addressed the limitations in the submission.

**Strengths And Weaknesses:**

Strengths:
1. The task of unsupervised keypoint localization for multiple instances is interesting and may facilitate future research.
2. The proposed method is technically sound and achieves comparable results with the state of the art.
3. The visualization of the prototype on the MNIST-Hard dataset is interesting. It is better to show the visualization of face/body keypoints too.

Weakness:
1. Multiple instances supporting is an important part of the proposed method, however, only a single instance quantitative evaluation is shown for some complex tasks (face/ body keypoint localization). The effectiveness of the proposed method needs to be further proved in such kinds of tasks.
2. The reconstruction loss L_recon seems to be mandatory in the proposed method and it has been performed for all the experiments.  But this loss may affect the robustness of the method on datasets with complex backgrounds which is more common in the multi-instance dataset. The authors may need to add more experiments on such kinds of datasets, e.g. the COCO dataset, The WIDER FACE dataset.
3. The details of how to learn the prototype is not clear in the submission. It is better to show how to update the prototype based on the sliced Wasserstein loss.
 4. The different settings of prototype number, i.e. M, need to be carefully analyzed.

---

### Official Review · Reviewer_tYeA · 2022-07-12

**Rating:** 5
**Confidence:** 3
**Soundness:** 2 fair
**Presentation:** 3 good
**Contribution:** 2 fair

**Summary:**

The following work proposes a formulation of the unsupervised latent-landmark learning autoencoder where the landmark bottleneck makes no assumptions about the number of instances of a keypoint to appear in each image -- an important distinction as prior approaches assume each landmark to appear exactly once per image. As with prior methods, two encoders are used, one for pose-invariant feature information, and the other for feature-invariant pose information. Output from the two encoders is combined and jointly decoded to reconstruct the original input. Pose invariance is achieved by feeding one encoder a thin-plate-spline warped image. Landmark activation locations are represented with predicted heatmaps. Unlike previous methods, these heatmaps are not assumed to have a single peak. Rather, the top-K activations are identified via NMS and retained for reconstruction. An online K-means loss is used to cluster feature representations at landmark locations where the resultant centroids are the resultant landmark descriptor.

**Questions:**

- It would be great if the authors could clarify for me what is meant by the intuition behind the Wasserstein loss (that the distributions of features and prototypes should match).
- Please address my concerns regarding the potential relationship to vector quantization literature.

**Limitations:**

limitations appropriately addressed

**Strengths And Weaknesses:**

Strengths
I think this work attempts to tackle an important issue, which is the applicability of unsupervised landmark methods. Due to the 1-instance-per-image assumption, these methods tend to be impractical when there are unknown number of instances or out-of-plane rotations.

Weaknesses
- Unless I've misunderstood a key component, the proposed method appears to be closely related to the autoencoders used in quantization literature, most notably VQVAE (van den Oord, 2017) and gumbel-softmax approaches. There's already a lot of extant literature based on top of that, learning discrete visual tokenizations of images that are critically neglected from this study. At the very least, these should serve as baselines for the clustering approach used in this work.
- There are some validity issues regarding the landmark-to-keypoint evaluation.
    - The first issue is directed towards the validity of the metric itself, and not specific to this work. Prior methods using this evaluation perform a linear regression without an intercept to map landmarks to keypoints. The lack of the intercept makes it such that the regression result is highly sensitive to the positioning of the origin in the coordinate space. On centered objects, placing the origin in the middle of the image will often result in better linear mappings than should the origin be placed in any of the 4 corners of the image.
    - The linear mapping formulation isn't really applicable to the landmark formulation in this method, as the authors noted in the text. It's not clear whether or not the linear layer used by this work includes the bias term. Furthermore, I'm not sure whether the tensor representation implies an ordering to the multiple instances of the same landmark, or this has been handled by the authors in their specific formulation.
   - In general, I don't think we can conclude anything from the comparison in table 3, though I understand the necessity of attempting the comparison regardless. Assuming my understanding is correct, this should probably be noted in the paper.

---

### Author Response · Authors · 2022-08-02
**Response to Reviewer tYeA**

**What is the relationship to autoencoders for quantization and the Gumbel softmax?**

We will include the discussion on VQVAE and Gumbel softmax in the related works section, and add vector quantization as a baseline for comparison.

In more detail, VQVAE and Gumbel softmax approaches are relevant, but are typically used for a different purpose: having a quantized latent space that can, for example, be easily translated by a transformer in order to generate images. We do not aim for a quantized latent space, but rather prototypes that **directly relate to semantics**.

This difference makes vector quantization a suboptimal choice when it comes to clustering, as shown in the table below, where we swap our clustering method with vector quantization (everything else remains the same) on MNIST HARD with 10 prototypes:

|  | Sliced Wasserstein  | Vector Quantization |
|---|---|---|
| Localization | 99.9% | 65.3% |
| Classification | 92.8% | 23.7% |
| Both | 92.8% | 15.3% |

Note how localization somewhat works, but classification completely fails, clearly indicating that vector quantization does not solve clustering. Qualitatively, we observe that some numbers get quantized into the same prototype, and others get split into two. This leads to detections also being somewhat “off” when estimating the center of the digits. We will add these results to Table 1 in the paper.

**Is the metric used in the landmark-to-keypoint evaluation valid?**

We believe it is. As the reviewer states, on datasets with roughly centered objects, placing the reference frame in the middle of the image may result in better linear mappings than placing it in a corner. Similarly, if the linear regressor has an intercept, it can learn to exploit these biases, even ignoring the input keypoints.

We train a linear regressor **without an intercept,** and place the origin on the **top left corner of the image,** the same as all baseline methods. The estimated landmarks are thus entirely dependent on the detected keypoints.

It is true that **the keypoints themselves** could exploit dataset biases such as objects being centered, but we evaluate on unaligned datasets, where the objects (e.g. faces in CelebA) are slightly misaligned: see L239-245.

**How we deal with multiple instances when converting to keypoints**

Our linear regressor cannot be identical to that used by previous methods because our approach can assign two or more keypoints to the same prototype. Our regressor needs more inputs: $2 \times K \times P$ ($K$: number of keypoints; $P$: number of prototypes), of which only $2 \times K$ are non-zero, the same as for the baselines (see L289-300). As the reviewer noticed, we forgot to mention that if multiple keypoints are assigned to the same prototype, we sort them **by their x coordinate from left to right** to fill the input tensor: we will add this detail to the paper.

As stated above, the frame of reference for the keypoints is the **top left corner of the image** and the regressor **does not have a bias,** as previous papers do.

**Can we conclude anything from the comparison in table 3?**

We use the same evaluation metric as all previous self-supervised keypoint learning papers, mapping keypoints to landmarks using a simple linear regressor without an intercept. Keypoints are in the same reference frame, with the origin at the top left corner of the image. Our regressor is essentially equivalent to that used by previous works, with a small modification to account for multiple instances. The comparison is thus meaningful. Our approach delivers comparable results while being applicable to multiple instances, unlike any of the baselines.

We believe we have answered the reviewer's questions regarding this evaluation, and kindly request clarification if that is not the case. We will reply within the reviewer-author discussion window.

**What is the intuition behind the Sliced Wasserstein loss?**

The intuition behind the loss is that the prototypes, if learned well, should resemble the data. In other words, their distributions should match. For example, should we have a perfect method for MNIST, the latent space should be 10 very narrow islands (almost a delta function), and each prototype should resemble each island. This is what we try to enforce on our prototypes via the sliced wasserstein loss, which minimizes the distance between prototype and feature distributions.

We show results on MNIST-hard. Without the sliced Wasserstein loss the prototypes collapse, as seen _[here](https://imgur.com/a/tXUHgg1)_: all features are assigned to a single prototype and the rest are not used.

|  | With SW loss | Without SW loss |
|---|---|---|
| Localization | 99.9% | 99.8% |
| Classification | 92.8% | 10.2% |
| Both | 92.8% | 10.2% |

---

### Author Response · Authors · 2022-08-02
**Response to Reviewer 9vPX**

**Quantitative evaluation on complex multiple instances**

We show that our method achieves comparable results to the SOTA on self-supervised landmark detection on the datasets typically used by the literature, and that it naturally generalizes to **multiple instances** (such as faces). We also show that by removing the single-object constraint our method can be applied to object discovery, generalizing to **multiple tasks.** While datasets like COCO/Wider Face contain multiple instances, they are also much more difficult and out of reach for the SOTA in fully unsupervised methods.

**Will the reconstruction loss work on images with more complex backgrounds?**

As we briefly discuss in the limitations section, our approach cannot deal with very complex backgrounds yet. This is also true for the SOTA in unsupervised methods, which are not ready for in-the-wild data. Note, however, that our approach performs well on images with different backgrounds, like those in unaligned-CelebA, and that unlike the SOTA in unsupervised learning, **it can deal with multiple instances,** which is itself a significant step towards getting such methods "out of the lab". We intend to explore this in future work.

**How are prototypes learned?**

Let us detail the process from L220-224. Within each training iteration, we first optimize encoder and decoder with fixed prototypes. We then optimize the prototypes with a fixed encoder/decoder. In Section A.2 (L534) we further describe how to sample prototypes to apply the sliced Wasserstein loss. Let us summarize this process:

${D}$: descriptors

$M$: number of descriptors

${P}$: prototypes

$N$: number of prototypes

**function** TrainPrototype (${D}$, ${P}$)
1. Calculate mixing ratio {$\pi_m$}
    * {${D}_m$} $\leftarrow$ divide ${D}$ into $M$ subsets where each subset is associated to a member in ${P}$ in terms of smallest $\ell_2$ norm
    * {$r_m$} $\leftarrow$ calculate the ratio of ${D}_m$ in ${D}$
    * {$\sigma_m$} $\leftarrow$ calculate variance of {${D}_m$}
    * $\sigma$ $\leftarrow$ calculate variance of {$\sigma_m$}
    * {$\alpha_m$} $\leftarrow$ {$r_m$} +$\sigma$
    * {$\pi_m$}  $\leftarrow$ {$\alpha_m$} /$\sum\alpha_m$
2. Sample from GMM
    * ${\tilde{P}}$ $\leftarrow$ initiate empty list
    * **for**  $p_m$ **in** ${P}$
      * append $\pi_m \times N$ samples from a Gaussian centered at $p_m$ with a predefined variance to ${\tilde{P}}$
3. Calculate sliced Wasserstein distance
    * $d$ $\leftarrow$ SW_distance(${\tilde{P}}$,${D}$)
 4. Train prototypes
    * Optimize ${P}$ by minimizing $d$

We will include this in the supplementary.

**How does the number of prototypes affect the results?**

Our method does not require the number of prototypes to be exact, but performs best when it is known in advance. In the table below we ablate the number of prototypes on MNIST-Hard. If it is smaller than the number of classes, each prototype learns to represent multiple classes, as shown _[here](https://imgur.com/a/n1bGe4V)_. With 5 prototypes representing 10 classes, classification accuracy will be at most 50%: we achieve 47.4%.  We can also evaluate the reverse mapping from class label to prototypes and check the consistency of assignments, which gives us 98.3% (this cannot be taken as accuracy!). If we have more prototypes than classes, prototypes learn to represent different modes of the class, as shown _[here](https://imgur.com/a/5UZbdTu)_. In general, a reasonable result can be achieved if the number of prototypes $P$ is larger than the number of classes. We will add these results to the paper.

|  | $P$=5 | $P$=5 (reverse) | $P$=10 | $P$=20 |
|---|---|---|---|---|
| Localization | 90.8% | 90.8% | 99.9% | 99.9% |
| Classification | 47.4% | 98.3% | 92.8% | 87.7% |
| Both | 43.0% | 89.2% | 92.8% | 87.7% |

**What happens if ${L_{recon}}$ or ${L_{sw}}$ are removed?**

The reconstruction loss is the main loss term used to train encoder and decoder. Our method will not work without it. All the other loss terms are used to regulate the latent space and encourage clustering.

Without the sliced Wasserstein loss, the network can easily learn a trivial solution where one prototype represents all the features and the rest are unused, as shown _[here](https://imgur.com/a/tXUHgg1)_. We also extend the ablation study below. We will add these results to the paper.

|  | With SW loss  | Without SW loss |
|---|---|---|
| Localization | 99.9% | 99.8% |
| Classification | 92.8% | 10.2% |
| Both | 92.8% | 10.2% |

**Why are localization results not affected by different settings of training losses?**

Localization is rather easy on MNIST-hard. Nonetheless we use this dataset to ablate as it is most “controllable”. The additional loss terms are designed to regulate the learned latent space to have better classification performance, as shown in Table 4.

**Typos**

Thank you for pointing them out. We will correct them.

---

### Author Response · Authors · 2022-08-02
**Response to reviewer q83N**

**Key concepts are well known, such as contrastive learning, clustering of feature representation, equivarience.**

We would like to emphasize that while the concepts that we utilize may exist in the literature, localized prototypes, especially ones that can support multiple instances, do not. It is non-trivial to build such a framework, and it is one of our main contributions.

We also note that our clustering loss based on the Sliced Wasserstein distance is novel and effective. As requested, we compared it against Vector Quantization (see first answer to reviewer tYeA) and k-means clustering with deep nets (see below), and both methods perform worse than ours.

**Test on more complex datasets (PASCAL3D+, VehiclePart, KITTI)**

We would like to note that the SOTA in unsupervised landmark (keypoint) learning is not ready for in-the-wild deployment. This includes our approach, which already takes a **significant step in this direction by removing the single-instance assumption**.

One of the limitations of our method is that we do not model the background explicitly, so that it struggles to work well on more complex datasets. As shown by the baseline methods for object discovery and landmark detection, self-supervised approaches only work with relatively simple datasets at the current stage. We noticed one concurrent work [F] published in ICLR'22  that extends the slot attention work to more complex images by using additional motion cues. However, the datasets used in [F] are still from synthetic images. As future work, we are planning to explicitly model the background and incorporate motion cues to let our method generalize to more complex images.

**Missing related works**

Thanks for bringing up these papers. We will add and discuss them.

[A, B] are early work on self-supervised landmark detection, and test on aligned-CelebA, where images are cropped to align the faces to the center of the image. As mentioned in L239, recent landmark detection methods perform really well on aligned-CelebA: we followed [H] to compare against most recent works on the more challenging unaligned-CelebA. Note that **[G], one of the baseline methods in our paper outperforms [A, B] in aligned-CelebA dataset by a large margin,** so it seems safe to assume that ours does too.

Regarding using simpler clustering methods, as both [C] and [E] utilize a form of k-means, we replace our clustering loss with a deep K-means clustering module [I] in order to have an apples-to-apples comparison. We report results in the table below: in short, it performs poorly. We suspect that this is due to the well-known downfall of K-means: it cannot recover from a degenerate state (multiple means being assigned to a single cluster). Our method, however, enforces that the distributions between prototypes and features match, and will thus be robust to this condition.

|  | K-means | Sliced Wasserstein |
|---|---|---|
| Localization | 99.5% | 99.9% |
| Classification | 21.1% | 92.8% |
| Both | 21.0% | 92.8% |

**Claims on modeling occlusion**

Unfortunately, the CLEVR dataset does not provide any annotation that can be used to evaluate occlusion quantitatively. We will tone down this claim to a qualitative insight, and include more qualitative results, such as _[these](https://imgur.com/a/5tV5bFr)_, in the supplementary material, to showcase more examples.

* [A] Unsupervised learning of object landmarks by factorized spatial embeddings
* [B] Unsupervised learning of object frames by dense equivariant image labelling
* [C] Visual concepts and compositional voting
* [D] Detecting semantic parts on partially occluded objects
* [E] Learning deep parsimonious representations
* [F] Conditional Object-Centric Learning from Video
* [G] Unsupervised Learning of Object Landmarks through Conditional Image Generation
* [H] Unsupervised Part Segmentation Through Disen- tangling Appearance and Shape
* [I] Deep k-Means: Jointly clustering with k-Means and learning representations

---

### Author Response · Authors · 2022-08-02
**Response to all**

We thank all reviewers for their input. We have addressed their questions below. Some replies required tables with results, which are listed inline, and figures, which have been uploaded to an external source while preserving anonymity (openreview does not support image uploads).

Please let us know if you have further questions during the reviewer-author discussion period.

---

> ### Author Response · Authors · 2022-08-08
> **Reminder that the discussion period ends on Tuesday, August 9**
>
> As the author-reviewer discussion period will end on Tuesday, we would like to know if our response answered the reviewers' concerns regarding the paper. Please let us know if you have any further questions, and we will do our best to reply by tomorrow.

---

### Meta-Review · Area_Chair_4hwt · 2022-08-24

**Recommendation:** Accept
**Confidence:** Certain

**Metareview:**

The meta reviewer has carefully read the paper, reviews, rebuttals, and discussions. The authors did a good job in rebuttal. The additional results and clarifications addressed the reviewers' concerns.  The manuscript crosses the acceptance bar. The authors are still suggested to revise the paper considering the reviewers' comments.

**Award:**

No

---

### Decision · Program_Chairs · 2022-09-14

Accept